# A Large-Scale Candidate-Gene Association Mapping for Drought Tolerance and Agronomic Traits in Sugarcane

**DOI:** 10.3390/ijms241612801

**Published:** 2023-08-15

**Authors:** Warodom Wirojsirasak, Patcharin Songsri, Nakorn Jongrungklang, Sithichoke Tangphatsornruang, Peeraya Klomsa-ard, Kittipat Ukoskit

**Affiliations:** 1Department of Biotechnology, Faculty of Science and Technology, Rangsit Campus, Thammasat University, Pathum Thani 12120, Thailand; warodomw@mitrphol.com; 2Mitr Phol Innovation and Research Center, Chaiyaphum 36110, Thailand; peerayak@mitrphol.com; 3Department of Agronomy, Faculty of Agriculture, Khon Kaen University, Khon Kaen 40002, Thailand; patcharinso@kku.ac.th (P.S.); nuntjo@kku.ac.th (N.J.); 4Northeast Thailand Cane and Sugar Research Center, Faculty of Agriculture, Khon Kaen University, Khon Kaen 40002, Thailand; 5National Center for Genetic Engineering and Biotechnology, National Science and Technology Development Agency, Pathum Thani 12120, Thailand; sithichoke.tan@nstda.or.th

**Keywords:** sugarcane, candidate-gene association study, target enrichment sequencing, drought tolerance

## Abstract

Dissection of the genetic loci controlling drought tolerance traits with a complex genetic inheritance is important for drought-tolerant sugarcane improvement. In this study, we conducted a large-scale candidate gene association study of 649 candidate genes in a sugarcane diversity panel to identify genetic variants underlying agronomic traits and drought tolerance indices evaluated in plant cane and ratoon cane under water-stressed (WS) and non-stressed (NS) environments. We identified 197 significant marker-trait associations (MTAs) in 141 candidate genes associated with 18 evaluated traits with the Bonferroni correction threshold (α = 0.05). Out of the total, 95 MTAs in 78 candidate genes and 62 MTAs in 58 candidate genes were detected under NS and WS conditions, respectively. Most MTAs were found only in specific water regimes and crop seasons. These MTAs explained 7.93–30.52% of phenotypic variation. Association mapping results revealed that 34, 59, and 104 MTAs involved physiological and molecular adaptation, phytohormone metabolism, and drought-inducible genes. They identified 19 pleiotropic genes associated with more than one trait and many genes related to drought tolerance indices. The genetic and genomic resources identified in this study will enable the combining of yield-related traits and sugar-related traits with agronomic value to optimize the yield of sugarcane cultivars grown under drought-stressed and non-stressed environments.

## 1. Introduction

Sugarcane (*Saccharum officinarum* L.), a member of the grass family (*Poaceae*), is an economically important crop grown globally and has been expanding due to an increase in demand for sucrose and bioenergy [1]. Each crop season is usually harvested at 8–12 months to reach a peak of sugar accumulation [2]. After the harvest in the first year as plant cane, the underground portion of the strikes gives rise to a succeeding crop known as ratoon cane [3]. Therefore, sugarcane experiences varying environmental factors affecting its growth and development in both plant cane and ratoon cane. Sugarcane is a relatively high water-demanding crop. Irrigation helps increase the sugarcane yields by around 23–54% as compared to the non-irrigated system [4], while droughts can lead to productivity losses of up to 60% in rainfed sugarcane fields, particularly during the formative growth phase [5]. The development of drought-tolerant sugarcane cultivars is critical to reducing climate-related risk, maintaining productivity, and enhancing the livelihood of sugarcane growers. In sugarcane, varietal improvements through a conventional breeding approach are challenging due to its genetic complexity and polyploid nature with significant levels of chromosomal mosaicism [6,7]. Genomics approaches for the identification of genes related to drought tolerance are of vital importance as a means of breeding new varieties of drought-tolerant sugarcane.

Effective drought response results from complex and dynamic physiological, morphological, biochemical, and molecular processes at cellular and systemic levels. Physiological and morphological responses of sugarcane plants vary according to the genotype, the duration and intensity of stress, and the type of tissue affected [8]. Under drought stress, higher productivity is related to higher stalk number, stalk height, and stalk weight. In contrast, stalk diameter is variable among varieties, being more dependent on the genotype than the environment [9]. For molecular processes, high throughput gene expression studies have focused on signal transduction and the role of phytohormones since water stress elicits extensive signal transduction networks, involving various transcription factors, protein kinases, and phosphatases [10,11].

Some studies have indicated that drought tolerance indices, such as the stress sensitivity index (SSI) and the stress tolerance index (STI) could provide useful information in association studies [12,13]. The drought-tolerant coefficient (DC) supplies a measure of drought effects based on the reduction of each trait under water stress conditions compared to well-watered conditions. It is therefore used for the identification of drought-tolerant genotypes. The membership function value of drought tolerance (MFVD) integrated multiple DC of different traits. It provided a comprehensive evaluation method for drought tolerance of materials based on the multi-indicator determination [14].

The expanding knowledge to understanding the genetic control of drought tolerance is of great importance for applying marker-assisted selection in the development of sugarcane cultivars with improved tolerance. The genetic variation underlying traits related to drought tolerance is polygenic, controlled by many loci with small and moderate effects [5]. A common approach for identifying genetic polymorphisms underlying adaptive traits is to test for associations between phenotypes and genetic variants [15,16]. Association studies are Genome-Wide Association Studies (GWAS) and candidate gene-based association studies [17]. GWAS have been performed in several crops, and many novel genes associated with economically important traits have been detected [18,19]. However, this approach requires numerous genome-wide markers, which are often noncoding or have unknown gene function [20]. The markers may not meet the threshold for statistical significance due to strict criteria after adjusting for multiple testing [21]. In contrast, candidate gene-based association studies use genetic variation in genes known or suspected to play a functional role in a phenotypic trait of interest [15,17,22]. This approach has appealing features, including reducing the substantial multiple testing penalties required for GWAS and directly connecting statistical testing with functional biological units [23,24]. Several studies have successfully identified associations between SNPs in candidate genes and phenotypic traits in plants such as sugarcane [25], rubber tree [26], wheat [27], and ryegrass [28].

In the present study, we performed large-scale candidate-gene association mapping based on candidate genes that have been predicted as related to physiological, morphological, and biochemical processes and, secondly, play a major role in cell signaling and gene regulation. The coding regions of 649 candidate genes were sequenced in a diverse collection of 159 sugarcane accessions using target enrichment sequencing. This study aimed to identify the candidate genes associated with sugar yield, agronomic traits, and drought tolerance indices, evaluated from plant cane and ratoon cane under irrigated and rainfed field experiments. We demonstrated that the large-scale candidate gene association approach could increase the chances of finding marker-trait associations (MTAs) that could affect the phenotypic traits related to drought tolerance in sugarcane.

## 2. Results

### 2.1. Phenotypic Traits Analyses

The accumulated monthly rainfall during the growing season, including the formative growth phase (December–March), was recorded for both non-stressed (NS) and water-stressed (WS) experiments (Appendix A). In both experiments, 42.86 mm. of water was applied for each irrigation. We applied eight irrigation times during the formative growth phase for the NS experiment to close crop water requirements in the dry season [29]. While only one irrigation time was applied in the WS condition to create proper conditions for measuring drought stress on the plants. Analysis of variance indicated that yield-related traits and sugar-related traits measured under the NS and WS experiments and in both plant cane and ratoon cane showed significant differences and wide variation among studied accessions (Figure 1). In plant cane, higher reduction in the WS in relation to the NS were observed for sugar yield (SY, 36.72%), cane yield (CY, 36.51%), single cane weight (SCW, 29.70%), cane length (CL, 22.59%), number of millable canes (NMC, 14.19%), internode length (IL, 12.06%), while cane diameter (CD, 2.01%), brix (BR, 0.12%), polarization (PO, −1.37%), purity (PU, −1.49%), fiber (FB, −2.13%), and commercial cane sugar (CCS, −2.35%) showed marginal variation. For ratoon cane, SY (61.83%), CY (49.62%), SCW (40.96%), CL (39.15), IL (36.84%), NMC (32.45%), CCS (22.11%), CD (16.25%), PO (15.03%), PU (10.57%), and BR (4.93%) were more adversely affected by drought, while FB (−14.47%) was increased in the WS experiment. The heritability (*h^2^*) for phenotypic traits of the plant cane varied from 70.42% (CL) to 93.54% (NMC) and 68.59% (PU) to 97.42% (NMC) estimated under the NS and the WS condition, respectively. For ratoon cane, *h^2^* varied from 69.78% (CY) to 91.90% (CD), estimated under the NS condition and from 75.36% (PU) to 94.19% (FB) estimated under the WS condition. The high *h^2^* indicated that the variations of the phenotypic traits were mostly due to genetic differences, and these traits were highly heritable.

Correlation coefficients (*r*) between certain stalk-related traits (CL, CD, IL, SCW, and NMC) and CY were positive and significant in both plant cane and ratoon cane evaluated under the WS experiment (Table 1 and Table 2). While CY had a positive and significant correlation with CL, CD, IL, and SCW but not with NMC in both plant cane and ratoon cane evaluated under the NS experiment. Positive and significant correlations were found between sugar-related traits (CCS, BR, PO, and PU) and SY in both plant and ratoon cane evaluated under the NS experiment. The correlations between sugar-related traits and SY evaluated under the WS experiment followed the same trend as in the NS experiment, except for BR, which was non-significant in either plant cane or ratoon cane. The negative correlations between FB and SY were significant in both plant and ratoon cane evaluated under both experiments. The ratooning ability (RA) evaluated in ratoon cane was highly and positively correlated (*p* < 0.01) with CL, NMC, CY, and SY in both experiments. Positive correlations among phenotypic traits between the NS and WS experiments were significant for both plant cane and ratoon cane. The exception was for the SY and RA, where the correlation was non-significant in ratoon cane.

### 2.2. Drought Indices Analysis

The variation of the drought-tolerant coefficient (DC) and the membership function value of drought tolerance (MFVD) was presented by means, standard deviations (SD), and ranges evaluated in plant cane and ratoon cane (Table 3). Cane yield showed the lowest drought-tolerant coefficient (DC_CY_) of 0.64 in plant cane, while the drought-tolerant coefficient of sugar yield (DC_SY_) was lowest (0.40) in ratoon cane, indicating that CY and SY were the most sensitive traits to water stress in plant cane and ratoon cane, respectively. Cane diameter showed the largest DC in both crop seasons, demonstrating that this trait was less affected by drought stress in this group of accessions. Mean of MFVD was 0.37 in the plant cane and 0.36 in the ratoon cane, ranging from 0.70 in plant cane and 0.67 in ratoon cane for the drought-tolerant genotype to 0.13 in the plant cane and 0.11 in the ratoon cane for the drought-susceptible genotype, classified according to Xu et al. (2023) [30]. Drought tolerant and susceptible accessions in the association population determined by the MFVD were presented in Appendix A.

Correlation coefficients between MFVD and DC of six yield-related traits (DC_CL_, DC_CD_, DC_IL_, DC_SCW_, DC_NMC_, and DC_CY_) and DC_SY_ of plant cane and ratoon cane were analysed (Table 4). The DC of all traits showed strong positive correlations (*p* < 0.01) with the MFVD in plant cane and ratoon cane. This indicated that these traits were closely related to the drought tolerance of sugarcane. The highest correlation in both crop seasons was found between DC_CY_ and MFVD. The DC of each trait and the MFVD evaluated in plant cane were significantly and positively correlated with those indices evaluated in ratoon cane except for DC_CD_.

### 2.3. Target Enrichment Sequencing and SNP/InDel Discovery

For target enrichment sequencing of the association population, 12,313 probes were designed to enrich 3293 exogenic regions of 649 candidate gene models according to the sugarcane monoploid genome (Appendix A). The designed probes covered 1.21 Mbp located on ten chromosomes of the reference genome, ranging from 375 probes of 18 genes (chromosome 5) to 2617 probes of 133 genes (chromosome 1). Sequencing the targeted regions of 159 accessions generated approximately 736.94 million reads in total, of which 662.35 million (89.88%) clean reads successfully mapped to the sugarcane monoploid genome in a range of 88.43% to 90.91% across the accessions. The number of clean reads per individual ranged from 2.61 to 6.01 million, with an average of 4.19 million reads. The read length ranged from 50 to 147 bp, averaging 136 bp. The average read depth was 396× in a range from 247× to 574×. Using the uniquely mapped reads, 187,044 SNPs and 42,834 InDels were identified in genic regions of the targeted genes, of which 180,763 (96.64%) and 37,703 (88.02%) were bi-allelic SNPs and InDels, respectively. After removing SNPs/InDels with ≥10% missing data and filtering for MAF ≥ 0.05, 31,764 markers, including 28,611 SNPs and 3153 InDels, were analyzed for linkage disequilibrium (LD). The squared allele frequency correlation (*r^2^*) for significant marker pairs (Bonferroni-corrected threshold α = 0.05) ranged from 0.14 to 1.00, with an average of 0.42 across ten chromosomes. When pairs of adjacent loci were found to demonstrate LD, only one of the two SNPs/InDels from each pair was used for marker-trait association (MTA) analysis, leaving 10,821 SNPs and 912 InDels in 649 candidate genes distributed on ten chromosomes of the sugarcane genome (Figure 2) with the average marker density of 18.19 markers/gene.

### 2.4. Population Structure

Population structure among the 159 sugarcane accessions was analyzed with an independent set of 5053 SNPs. Using the independent markers can avoid dependency among terms in the model and prevent the structure from absorbing the QTL effects from the model [31]. The association population structure was studied using the discriminant analysis of principal components (DAPC) method. Following the analysis of the Bayesian Information Criterion (BIC) profile and using the ‘a-score’ criterion, the optimal number of clusters was predefined to 3 (Figure 3a), and the optimal number of principal components (PCs) was set to 6 (Figure 3b) which explained the cumulative percentage variance of 16.8%. Therefore, these 6 PCs and 3 clusters were used for DAPC. The distribution of individuals into the 3 clusters is represented along the first two axes of DAPC (Figure 3c). Cluster 1 (green circle) comprised 54 accessions, mainly (48 accessions) from exotic introductions (Appendix A). Cluster 2 (red circle) was a large group comprising 62 accessions, including 17 accessions of Thai cultivars, 26 accessions of breeding lines which shared a pedigree, and 19 accessions from exotic origins. Cluster 3 (blue circle) consisted of 38 accessions. The majority of accessions were Thai cultivars (11) and breeding lines (20), while the remaining seven accessions were exotic origins. The remaining five accessions, including Thai cultivars (2) and breeding lines (3), did not have clear group characteristics and were considered admixed accessions (Appendix A).

### 2.5. Association Mapping

A total of 197 marker-trait associations (MTAs) in 141 candidate genes were identified using Fixed and random model Circulating Probability Unification (Farm CPU) based on Bonferroni-corrected threshold α = 0.05 (*p*-value ≤ 4.26 × 10^–6^) for all evaluated traits. In total, 95 (48.22%) MTAs in 78 candidate genes and 62 (31.47%) MTAs in 58 candidate genes were detected under NS and WS conditions for yield-related traits, sugar-related traits, and RA. While 40 MTAs in 34 candidate genes were identified for drought tolerance indices. These QTLs explained 7.93–30.52% of phenotypic variation. Manhattan plots and quantile-quantile (QQ) plots were drawn to represent MTAs for each phenotypic trait of plant and ratoon cane evaluated under NS and WS conditions (Appendix A) and for drought tolerance indices of plant and ratoon cane (Appendix A).

#### 2.5.1. Marker-Trait Association Under NS Condition

Under non-stressed conditions, 50 MTAs in 45 candidate genes were identified for yield-related traits, including CL (8), CD (8), IL (3), SCW (10), NMC (12), CY (2), and SY (7), while 44 MTAs in 38 candidate genes were observed for sugar-related traits including CCS (10), FB (7), BR (7), PO (10), and PU (10) (Figure 2). One marker, an InDel on chromosome 9 (InDel02945), was found to significantly associate with RA (*p*-value = 8.23 × 10^–6^). The percent of phenotypic variation explained (PVE) ranged from 7.93% in FB of plant cane to 24.02% in the CD of ratoon cane (Appendix A). For SY, which is the final product of the sugar industry, three markers, including SNP08052, SNP20355, and SNP27968 and four markers, including SNP07858, SNP20008, SNP25403, and SNP32155, showed significant association in plant cane and ratoon cane, respectively. The phenotypic variation explained by the markers associated with SY was relatively high, between 9.96% (SNP20355) to 19.65% (SNP07858) (Appendix A). Most of the MTAs identified under NS condition were crop-season specific, suggesting the influence of genotype × crop season interaction on the phenotype of the traits measured in plant and ratoon cane. However, SNP05953 was associated with CD in both crop seasons. Pleiotropic markers having an association with more than one trait were also found. In particular, SNP05953 was found to be significantly associated with CD, SCW, and NMC, while SNP06940, SNP17531, SNP23693, SNP25403, and SNP28774 were significantly associated with more than one sugar-related trait. However, their strength of association differs depending on traits, water regime and crop season.

#### 2.5.2. Marker-Trait Association Under WS Condition

Under water-stressed conditions, 62 MTAs in 58 candidate genes were detected for CL (5), CD (12), IL (3), SCW (3), NMC (1), SY (5), CCS (8), FB (6), BR (6), PO (8), and PU (5) (Figure 2). No significant association was observed for CY and RA. Associated markers of these traits showed phenotypic variation explained from 8.20% to 22.38% (Appendix A). The highest phenotypic variation (22.38%) was explained by SNP32494 associating with FB. For yield-related traits, maximum phenotypic variation (19.98%) was explained by SNP30896 significantly associated with CD. For SY, five markers, including SNP11186, SNP12296, SNP17009, SNP28936, and InDel01261, showed significant association in plant cane. The phenotypic variation explained by the markers associated with SY ranged from 9.22% (SNP11186) to 18.56% (SNP17009). The pleiotropic marker, SNP28798, was associated with CD and SCW. Similar to the NS condition, most MTAs were significant only in a specific water regime and crop season except SNP22843, SNP24679, SNP28798, and SNP32494, which were associated with BR, CD, SCW, and FB, respectively, in both NS and WS conditions.

#### 2.5.3. Marker-Trait Association for Drought Tolerance Indices

A total of 40 significant MTAs in 34 candidate genes were detected for drought tolerance indices, and most MTAs (10 MTAs) were clustered on chromosome 1 (Figure 2). The average phenotypic variation explained was 14.19% and ranged from 9.46 to 30.52%. The most significant MTA was on chromosome 1 associated with DC_SCW_ and explained the highest phenotypic variation. A total of 24 of the significant MTAs detected were associated with two drought indices (Appendix A). DC_SY_ and DC_SCW_ had the highest number of significant associations, each with 15 and 9 MTAs explaining the average of 14.36% and 16.44% phenotypic variation, respectively. Three pleiotropic markers, including SNP01310, SNP03302, and SNP17032, were identified for drought tolerance indices. The significant MTAs for MFVD, which was considered on average DC indices of yield-related traits, were detected in both plant cane, including SNP17032, SNP25057, and SNP28494, and ratoon cane, including SNP03302, SNP16162, and SNP21960. The most significant association with MFVD was SNP21960, located on chromosome 6 and explained 15.74% phenotypic variation. The Manhattan plots for MTAs can be found in Appendix A.

## 3. Discussion

### 3.1. Phenotypic Data Analysis

Drought simultaneously affects a varied number of morphological and physiological traits in plants. A single characteristic cannot reflect the complex traits of the drought tolerance mechanism, so more traits should be considered to evaluate drought tolerance. In the present study, the results indicated that drought during the formative phase reduced SY, CY, SCW, CL, NMC, and IL at harvest in both plant cane and ratoon cane. Water stress has been reported to reduce stalk number, stalk height, stalk weight, and number of millable stalks and ultimately cause a decline in cane yield [9,32]. In the plant cane, the marginal reduction in CD due to drought stress was more dependent on the genotype than the environment. This result indicated less influence of water stress on stalk diameter, in agreement with previous reports [9]. In sugarcane, breeders must select traits positively correlated with cane yield. In the present study under the WS condition, positive and significant correlations between certain stalk-related traits (CL, CD, IL, and SCW) and CY were similar to the NS condition in both plant cane (Table 1) and ratoon cane (Table 2), except for the correlation between NMC and CY, which had a highly significant correlation (*p*-value < 0.01) in both plant cane (*r* = 0.50) and ratoon cane (*r* = 0.77) evaluated under the WS experiment but not under the NS experiment. These findings suggested that the selection of sugarcane genotypes for drought tolerance at the early growth phases with higher productivity associated with cane yield should be made with special attention to higher stalk number, stalk height, and stalk weight. This result is in agreement with previous reports [9,33]. However, the parameters of yield-related traits are involved in plant development and are essential in yield determination under water-stressed conditions. Marker trait associations discovered for SCW, CL, IL, NMC, including CD, might enhance cane yield, mainly because these traits were strongly correlated with cane yield (Table 1 and Table 2).

As observed in plant cane, sugar-related traits (BR, PU, PO, FB, and CCS) were less drought-sensitive. This is probably because the drought condition was imposed during the stool formation period, i.e., of intense growth. Ramesh and Mahadevaswamy (2000) [34] reported that the effect of water stress on sucrose accumulation is observed later in the season, i.e., between 240 and 360 days after planting. For ratoon cane, both yield-related traits (CY, SY, SCW, CL, IL, NMC, and CD) and sugar-related traits (CCS, BR, PU, and PO) were more affected by drought. In this study, irrigation was not applied to the ratoon cane in the WS experiment, and therefore the germination of the ratoon cane was solely dependent on stored soil water. A low yield of the ratoon crop is mainly due to the low ratooning ability of sugarcane varieties and the effect of drought on the survival of the stool and subsequent growth of the ratoon crop [35]. In this study, the ratooning ability was found to vary among sugarcane accessions (Figure 1). This variation has a genetic basis and potential for breeding sugarcane varieties with strong ratooning ability. MTAs identified for ratooning ability might potentiate cane yield and sugar yield in ratoon crops, because RA was strongly correlated (*r* > 0.5) with CY and SY in ratoon crops evaluated under drought conditions (Table 2).

Drought indices provide a measure of drought effects based on the reduction of each trait under water-stressed conditions compared to non-stressed conditions and therefore are useful for screening drought-tolerant genotypes. Genotypes with high values for drought indices are generally regarded as stable under stressed and non-stressed conditions. In the present study, MFVD was significantly and positively correlated with the DC of all yield-related traits in both plant cane and ratoon cane. In a previous study, positive correlations have been found between MFVD and DC of biomass, stalk weight, stalk diameter, and stalk height at the growth stage and seedling stage [30], which is also supported by our results. In the association mapping panel evaluated in this study, we observed a wide variation of DC and MFVD, suggesting that there are genetic sources of drought tolerance in the panel used in this study. The genotypes that showed the least decrease in the DC and MFVD values are more likely to be drought tolerant. The MFVD, which combined the DC of physiological or morphological traits that could be easily estimated, was used to investigate the sugarcane drought tolerance in this study. According to MFVD evaluated in plant cane (Appendix A), the most drought-tolerant cultivar (MFVD > 7.0) was from South Africa (NCo382). The following most drought-tolerant cultivars (0.6 ≤ MFVD ≤ 7.0) were three Thai cultivars (Biotec 1, Biotec 3, and Biotec 6) which were related in pedigree. For MFVD evaluated in ratoon cane, seven Thai breeding accessions (related in pedigree) and one accession from the USA were classified as drought tolerant. This result suggested several drought-tolerant Thai cultivars and breeding lines have been adapted to rainfed environments and could be recommended as donors for sugarcane drought-tolerant improvement breeding. Contrasting genotypes identified in the present study may serve as promising material to develop mapping populations for further genetic dissection of the trait.

### 3.2. SNP Detection

Discovering DNA polymorphisms in highly polyploid genome species remains challenging due to their large genome size and high heterozygosity. To improve the detection of sequence variations for these species, it was suggested to enrich a subset of the genome to reduce genome complexity and increase the sequencing depth for accurately calling genotypes [36,37]. Therefore, target enrichment methods that selectively capture genomic DNA regions of interest before sequencing are widely applied [38,39,40]. In this study, the target enrichment sequencing approach was applied in the association population panel for deep genotyping of the exogenic regions of 649 candidate genes to obtain an average sequencing depth of 396×. At this sequencing depth, it could achieve the minimum sequence coverage (108×) for capturing all possible genotypes of the sequence variants in sugarcane with up to 12 sets of chromosomes [41] and ensure that single-dose SNPs were not called homozygotes. Moreover, regions of interest with either low or multi-allelic genotyping or a high percentage of artefact (off-target) reads were excluded. These ensured that almost all the targeted regions had enough read to allow high-quality genotype assignment and comprehensively detect adaptive genetic variations.

### 3.3. Population Structure Analysis

Population structure in an association population might be a confounding factor and must be addressed to avoid false associations. Discriminant analysis of principal components (DAPC) divided the association population into three clusters (Figure 3c). The study’s use of genotypes from exotic introductions might be the reason for cluster 1 (Figure 3c) in the association mapping panel. Sugarcane accessions with different origins sometimes clustered together, suggesting shared alleles, while accessions from the same country were placed in different groups [42]. This result might be explained by germplasm exchange among breeding programs around the globe. Overall, Thai breeding lines and cultivars in clusters 2 and 3 were in accordance with the pedigree developed from common founding parents used in their crossing plan. The use of elite breeding material for association mapping may reduce the number of significant SNPs. However, utilising Thai cultivars and exotic introductions will help explore untouched parts of the genome having minor effects on the target traits.

### 3.4. Association Mapping

Drought tolerance is affected by complex traits that are influenced by several genes and have a complex genetic inheritance. With the advent of target enrichment sequencing, we conducted a large-scale candidate gene association study of 649 candidate genes in a sugarcane diversity panel to identify genetic variants underlying agronomic traits and drought tolerance indices evaluated under drought-stressed and non-stressed environments. This approach can discover 197 significant MTAs (Figure 2) in 141 candidate genes associated with the 18 evaluated traits using a stringent significance threshold. The phenotypic variation explained by these SNPs/Indels was relatively high (7.93–30.52%). Several reasons could explain these results. First, candidate genes that contribute to stress response were specifically selected for this study, increasing the chances of finding MTAs that could affect the phenotypic traits. Second, controlled experimental conditions could heighten differences in response to drought and, thus, increase the proportion of phenotypic variation explained by the SNPs/InDels. However, it cannot be ruled out that the small population size used in our study could affect the percentage of phenotypic variation explained by the SNPs/InDels. Studies with small sample sizes could be affected by the Beavis effect [43], resulting in an overestimation of marker-trait associations in QTL or association studies.

Although previous reports indicated that the selected candidate genes in this study were involved in drought stress response, MTAs were detected under both NS and WS conditions. It is possible that even the trial grown under the NS condition experienced some degree of moisture stress due to the uneven distribution of monthly rainfall (Appendix A) after the formative growth phase, thereby inducing the expression of the evaluated genes. In addition, some genes enhance drought tolerance while maintaining normal growth under unstressed conditions [44]. In the NS condition, significant MTAs were identified in 46 candidate genes involved in hormone signaling, transcriptional regulation and sugar metabolism. Pleiotropic genes showing significant MTAs with yield-related traits and sugar-related traits are helpful in marker-assisted selection as they may play a vital role in increasing QTL pyramiding efficiency [45]. For yield-related traits, the pleiotropic SNP05953 at the Sh01_g040220 gene explained high PVE (9.63–24.02%) significantly associated with CD in crop season with SCW and NMC in plant cane. The Sh01_g040220 gene is annotated as Dof-type zinc finger protein or Dof transcription factor. In Arabidopsis, the cambium and radial organ enlargement activity is associated with plant biomass. Loss-of-function of CYCLIN D3;3 (CYCD3;3), which was directly bound and regulated by Dof3.4, was found to cause a reduction in radian organ size, including stem and hypocotyl diameter linked to reduced cambial cell proliferation [46]. This result is similar to Chen et al. (2020), who found that Dof54 overexpression lines of apple plants had higher stem diameter but not plant height under long-term drought stress than control lines [47].

Essential pleiotropic candidate genes identified for sugar-related traits in NS condition involved four candidate genes. The SNP, SNP06940, at the Sh02_g004260 gene, annotated as prolyl 4-hydroxylase (*P4H*), was significantly associated with CCS and PU. It was suggested that the *P4H* genes were involved in fruit ripening because some *P4H* genes in tomatoes were not expressed in immature fruit [48], while some *P4H* genes in kiwifruit and banana were upregulated by ethylene-induced ripening [49,50]. The *P4H* might be involved in sucrose accumulation in sugarcane by inducing ethylene production during the maturity stage [51]. The SNP, SNP17531, associated with CCS and PO, was located on the Sh04_g023500 gene that encodes indole-3-acetic acid (IAA)-induced protein ARG7. This gene is induced by IAA and is closely related to the drought resistance of white clover [52]. The regulation of *ARG7* was proposed to involve sugar content [53]. The SNP, SNP25403, at the Sh08_g000220 gene, annotated as E3 SUMO-protein ligase SIZ2, was associated with CCS, BR, and SY. SUMO (Small Ubiquitin-like Modifier) proteins are a family of small proteins covalently attached to and detached from other proteins in cells to modify their function. Suppression of the *SIZ1* gene resulted in increased expression of genes related to sucrose and starch degradation [54] and reduced levels of sucrose and other sugars such as glucose, fructose, xylose, and maltose. Another essential candidate gene associated with BR and CCS was the Sh09_g010250 gene, containing SNP28774, and annotated as a glycosyltransferase (GT). UDP-glycosyltransferase is the largest glycosyltransferase family in plant species. UDP-glycosyltransferase is thought to be involved in the transfer of glycosyl residues from activated nucleotide sugars to acceptor molecules. In sugarcane, UDP-glycosyltransferase was detected in QTL I-49 for brix, pol, purity and sugar yield by our previous work [42]. This study identified an InDel, InDel02945, associated with RA. This marker was located on the Sh04_g022390 gene, annotated as nuclear transcription factor Y (NF-Y) subunit B. The NF-Y complex regulates the expression of target genes by directly binding the promoter CCAAT box or by physical interaction and mediating the binding of a transcriptional activator or inhibitor. It was revealed that NF-Y executes different biological functions, including flowering [55], fruit ripening [56] and response to adverse environments [57], in the processes of plant growth and development.

The MTAs for yield-related traits and drought tolerance indices identified exclusively under drought conditions could play a vital role in drought tolerance. They would significantly contribute to the development of molecular tools for marker-assisted selection and identification of genes of interest. The present study identified three interesting candidate genes showing pleiotropic effects for yield-related traits and drought tolerance indices. First, SNP17009 and SNP17032 were identified in the Sh04_g022390 gene and associated with NMC, SY, DC_SY_, and MFVD. This gene is annotated as auxin-responsive protein SAUR32. The SAUR gene family are growth factors that are essential for both normal plant development and adaptation to environmental conditions, acting with or without auxin [58,59]. Second, InDel03511, associated with IL and DC_IL_, was located on the Sh10_g017200 gene, annotated as Phosphatase DCR2 (dosage-dependent cell cycle regulator 2). The DCR2 gene played a positive role in cell cycle progression and stress response, acting as an antagonist of unfolded protein response (UPR) [60,61]. Third, SNP01278 and SNP01310 were associated with three drought indices, DC_SCW_, DC_CY_ and DC_SY_, and explained high phenotypic variation (13.98–17.79%). These two SNPs were located on the Sh01_g008050 gene, annotated as 1-aminoacyclopropane 1-carboxylate synthase (ACC) synthase, the key enzyme in ethylene biosynthesis [62]. In plants, ACC synthase knockout reduced ethylene content and delayed leaf senescence under water stress, enhancing chlorophyll activity and stomatal conductance and positively contributing to drought tolerance [63].

In many cases, genes that influence a particular trait under stress conditions also control the trait under non-stressed conditions [64]. The effects of such loci may not be influenced by the change in the external environment. Such genomic regions could be helpful in gene introgression when breeding for broad adaptation. In the present study, three MTAs, including SNP32494, SNP28798, and SNP22843, were commonly detected across different conditions. First, SNP32494 was located on the Sh10_g011820 gene and associated with FB. This gene is annotated as EREBP-4-like protein (ethylene-responsive element-binding factor 4 (EREBP-4) or ethylene-responsive transcription factor 4 (ERF4)). It has been observed that the overexpression of *ERF4* from *Brissica rapa* increases tolerance to salt and drought stresses in *Arabidopsis* [65]. Additionally, it has been suggested that the reduction or down-regulation of ERF4-like protein promotes fiber development and secondary wall formation and increases stem biomass [66,67]. Second, SNP28798, associated with SCW, was located on the Sh09_g010250 gene that encodes glycosyltransferase. The function of glycosyltransferase under the NS condition was described in the previous discussion section. In drought stress conditions, the glycosyltransferase encoding gene has been suggested that this gene has a major effect on drought stress in spring wheat [68] and also contributes to abiotic stress tolerance in Arabidopsis [69], maize [70] and rice [71]. Third, SNP22843 is located on the Sh06_g020100 gene annotated as DNA mismatch repair protein MSH3. The MSH3 is involved in correcting DNA replication errors and maintaining the genome integrity in plants under environmental stresses [72]. The overexpression of DNA repair genes in transgenic plants makes these plants more tolerant of drought, salinity, and other stress conditions [73,74].

## 4. Materials and Methods 

### 4.1. Plant Materials and Experimental Design

The association population comprised a diverse collection of 159 sugarcane accessions derived from Argentina (2), Australia (13), Barbados (4), Brazil (3), China (2), Cuba (1), Fiji (7), Guyana (1), India (6), Mauritius (4), Philippines (3), Reunion (1), South Africa (3), Sri Lanka (1), Taiwan (10), USA (9), Thai breeding lines (52), Thai varieties (33), and four of unknown origin (Appendix A). An irrigated (non-stressed, NS) and a rainfed experiment (water stress, WS) were conducted in plant cane (December 2017–December 2018) and in first ratoon cane (December 2018–December 2019). Accumulated monthly rainfall in the growing season was recorded for both experiments (Appendix A). The NS experiment was conducted at Mitr Phol Innovation and Research Centre (sandy clay loam soil), Phu Khieo, Chaiyaphum (16°26′49.1″ N 102°08′06.9″ E). Full irrigation was imposed to minimize the level of water stress as much as possible by drip irrigation (42.86 mm. applied per each irrigation) given every 15 days from December to March (8 irrigation times). To determine the effect of drought during the formative growth phase (60–150 days after planting), the WS experiment was conducted at Agronomy Field Crop Station (sandy loam soil), Faculty of Agriculture, Khon Kean University (16°28′39.7″ N 102°48′32.2″ E). This experiment received only one irrigation at the beginning of the crop cycle to establish the crop. The two experiments received rainfall from summer to the rainy season (April–September). The experiments were laid out in a randomized complete block design (RCBD) with two replicates in three-row plots with 5 m in length and plant spacing of 1.65 m between rows and 0.5 m between plants, maintaining ten plants per row. Standard commercial cultivation practices were carried out according to the recommended crop production practices.

### 4.2. Phenotyping and Field Data Analysis

This study aimed to assess the drought stress responses of yield-related traits and sugar-related traits of sugarcane cultivars grown under drought-stressed and non-stressed environments. Phenotypic data was measured during the maturity phase (360 days after planting) from six main random stalks from one middle row of each plot. Sugar-related traits, including brix (BR), polarization (PO), purity (PU), commercial cane sugar (CCS) and fiber (FB) were collected. BR is an estimate of the total dissolved solids in the juice and was measured with a hand-held refractometer from a sampling punch taken at half-height on the stalk. PO was estimated by polarization and calculated as grams of sucrose per 100 g of sugar. PU was calculated as a percentage of pure sucrose in dry matter [42]. After juice extraction, the fresh and dry weight of the remaining stalks was weighed for calculating FB. CCS was measured and calculated as a standard function of BR, PO, and FB. For yield-related traits, data on the number of cane length (CL), cane diameter (CD), internode length (IL), single cane weight (SCW), number of millable canes (NMC), and cane yield (CY) were recorded during December for both plant cane and ratoon cane. All numbers of stalks within the harvested area were recorded as NMC. The weight of all millable canes was measured as CY. Sugar yield (SY) was calculated from (CY × CCS)/100. In this study, we only focused on ratooning ability (RA) of the first ratoon crop, the most widely used cane crop for commercial production. Ratooning ability was calculated from (CY of ratoon cane/CY of plant cane) × 100. The drought-tolerant coefficient (DC) of yield-related traits measured was calculated as:(1)$${DC}_{ij}=\frac{X_{ijWS}}{X_{ijNS}}$$
where DC_ij_ was the drought-tolerant coefficient of the trait (j) for the genotype (i); X_ijWS_ and X_ijNS_ were the values of the trait (j) for the genotype (i) evaluated under WS and NS conditions, respectively [30]. Drought tolerance for each sugarcane cultivar was evaluated by the membership function value of drought tolerance (MFVD) using the approach of Chen et al. (2012) [75]. According to the DC, the MFVD was calculated as follows:(2)$$U_{ij}=\frac{{DC}_{ij}-{DC}_{ijmin}}{{DC}_{jmax}-{DC}_{jmin}};U_{i}=\frac{1}{n}\sum_{j-1}^{n}U_{ij}$$
where U_ij_ is the membership function value of the trait (j) for the cultivar (i) for drought tolerance; DC_jmax_ is the maximum value of the drought-tolerant coefficient for the trait (j); DC_jmin_ is the minimum value of DC_ij_. U_i_ (MFVD) was the average value of the membership function of all the traits for the cultivar (i) for drought tolerance. The outlier was removed for some traits. Traits that were distributed normally with the Shapiro–Wilk test was considered to be normal data (Appendix A). The experimental data were subjected to analysis of variance (ANOVA). They determined a significant difference between means based on the least significant difference (LSD) test at 0.05 probability level using Statistix 10 software program (Analytical Software, Tallahassee, FL, USA). Pearson’s correlation coefficients (*r*) were determined using the R programming language [76]. The genetic variance and error variance from the ANOVA were used to estimate the broad sense heritability (*h^2^*) for each trait. The *h^2^* was estimated based on each environment according to the following equation:(3)$$h^{2}=\frac{\sigma_{g}^{2}}{\left(\sigma_{g}^{2}+\frac{\sigma_{e}^{2}}{r}\right)}$$
where $\sigma_{g}^{2}$ is the genetic variance, $\sigma_{e}^{2}\;$ is the residual variance, *r* is the number of replicates.

### 4.3. Candidate Genes and SNP/InDel Discovery

Candidate gene association mapping was performed using SNP/Indel markers. Total genomic DNA was extracted from leaf tissue using the cetyltrimethyl ammonium bromide (CTAB) method, as described by Gawel and Jarret [77]. The quality and quantity of the total DNA were determined using 1% agarose gel electrophoresis and spectrophotometric measurement, respectively. For SNP/InDel discovery, the target enrichment sequencing was performed on 649 selected candidate genes (Appendix A) involved in physiological and molecular adaptation (117 genes) [78,79,80], phytohormone metabolism (183) [81] and drought inducible genes (349 genes) [11,82,83]. Probes of 120-bp oligonucleotides for the candidate genes were designed to capture all exon regions of the candidate genes by blasting the genes against the sugarcane monoploid genome [6]. The enriched libraries were sequenced by Illumina HiSeq 4000, and the raw reads were trimmed using Trimommatic version 0.39 [84] with a PHRED score ≥ 20. Trimmed clean reads with lengths ≥ 50 were mapped on the sugarcane monoploid genome using BWA-men [85]. Based on uniquely mapped reads with a mapping quality score ≥ 30 and base quality score ≥ 20, SNPs and InDels were called by Unified Genotyper implemented in Genome Analysis Tool Kit (GATK) v4.1.3.0 [86]. All the settings for Trimommatic, BWA-mem, and GATK were default. Only SNP and InDel loci with at least 56 reads were kept. SNPs and InDel with missing data ≥ 10% and minor allele frequency (MAF) < 0.05 were discarded. The remaining SNP/InDel markers were used to analyse LD and marker-trait associations.

### 4.4. Population Structure and Relative Kinship

To control for confounding effects in association mapping, the population structure and relative kinship were investigated based on an independent set of 5053 polymorphic SNPs evenly distributed across all ten sugarcane chromosomes (Appendix A). Since sugarcane is polyploid, outcrossing, and clonally propagated, we applied a discriminant analysis of principal components (DAPC), implemented in the adegenet R package [87,88], to capture population structures in this sugarcane association population. This method partitioned genetic variation into within and among group components without assumptions on linkage disequilibrium or Hardy–Weinberg equilibrium [88], which aims at identifying and describing clusters of individuals without prior knowledge of groups. Firstly, K-mean was run sequentially from 1 to 40 genetic clusters (K) to determine the best number of clusters using data from the principal coordinates (PCs). The most likely number of clusters was chosen based on the lowest Bayesian Information Criterion (BIC). Then, the optimal number of PCs to be retained was performed through maximization of the ‘*a*-score’ criterion that is associated with the lowest root mean square error [88]. Finally, DAPC was performed considering the most likely number of clusters (k) and the optimal number of identified PCs. The clusters indicated by DAPC were plotted in a scatterplot considering the first and second linear discriminants. The pairwise kinship coefficients were calculated with the function “centered IBS” (identity by state) implemented in TASSEL 5 (Trait Analysis by Association, Evolution and Linkage) software [89]. Negative values between individuals were set to 0, which means they have a weaker relationship than random individuals [90]. The two first coordinates of DAPC results (Q matrix) and kinship matrix (K matrix) were used as covariates for subsequent association analyses.

### 4.5. Linkage Disequilibrium and Association Mapping

Linkage disequilibrium (LD) was assessed from the squared allele frequency correlation (*r*^2^) among all SNP/InDel markers of each chromosome using the software package TASSEL 5 [89]. Bonferroni correction was used as a threshold to determine LD extent by dividing the significance threshold of 5% (type I errors) by the number of comparisons. When pairs of adjacent loci demonstrated strong LD, one of the two markers from each pair was used for marker-trait association analysis. Associations between SNP/InDel markers and the phenotypic traits were analyzed using Fixed and random model Circulating Probability Unification (FarmCPU) [91] integrated tool (GAPIT) in R with default parameters. The correction for population stratification and cryptic relatedness was performed by employing the coefficient of co-ancestry kinship and the two first coordinates of DAPC as covariates by looking at the model fit using Quantile-Quantile (QQ) plots generated using the ‘qqman’ package in R. The threshold to declare a significant association was Bonferroni-corrected significance threshold α = 0.05 (*p*-value ≤ 4.26 × 10^−6^). Manhattan plots were drawn to represent marker-trait associations. The percentage of phenotypic variance explained (PVE) by associated SNP/InDel markers was calculated following Teslovich et al. (2010) [92].

## 5. Conclusions

To our knowledge, this is the largest and most comprehensive candidate gene association mapping for drought tolerance in sugarcane. The large-scale candidate gene approach offers the advantage that highly relevant genes can be prioritized and tested and can discover 197 significant MTAs (Figure 2) in 141 candidate genes associated with 18 evaluated traits with the Bonferroni correction threshold (α = 0.05). Association mapping results revealed 19 pleiotropic genes related to more than one trait and many genes associated with DC and MFVD index regarding stability under stressed and non-stressed conditions. The genetic and genomic resources identified in the present study will enable the combining of yield-related traits and sugar-related traits with agronomic value to optimize the yield of sugarcane cultivars grown under drought-stressed and non-stressed environments.

## Figures and Tables

**Figure 1 ijms-24-12801-f001:**
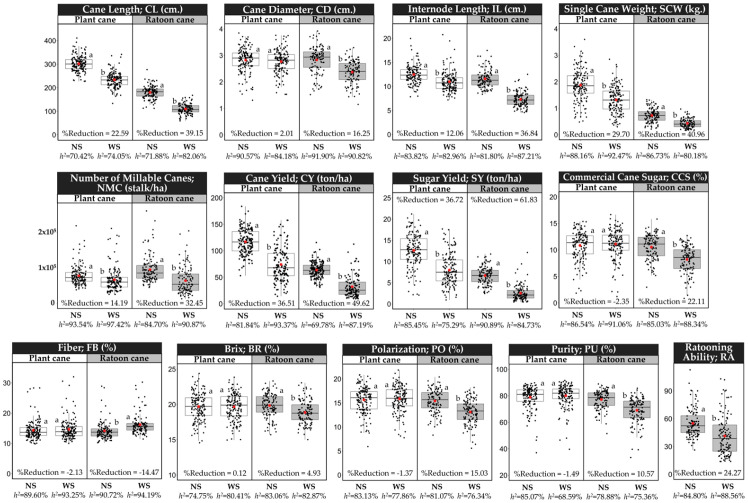
Box plots showing the descriptive statistics of the phenotypic traits measured under non-stressed (NS) and water-stressed (WS) conditions. Different letters on the boxes indicate a significant difference by the least significant difference (LSD) test at *p* < 0.05. The box’s horizontal line and red circle represent the median and the mean, respectively. The lower and upper limit of the box, lower and upper whisker, represents Q1 (25th percentile), Q3 (75th percentile), (Q1–1.5IQR) and (Q3 + 1.5IQR), respectively. IQR—interquartile range. The black color dots on the boxes indicate the distribution of the 159 sugarcane accessions.

**Figure 2 ijms-24-12801-f002:**
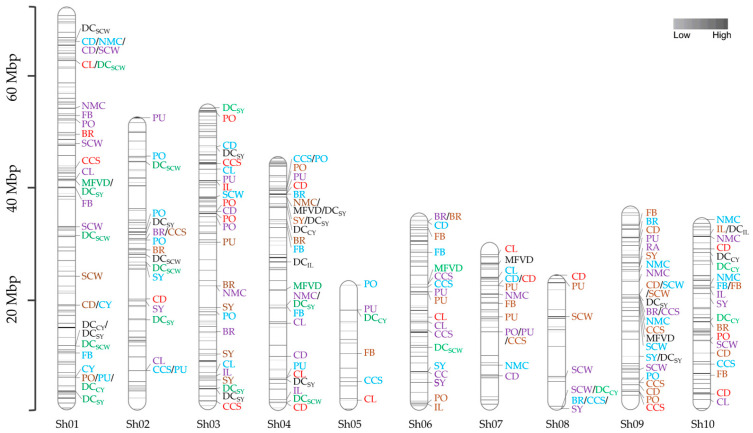
Chromosomal locations of 11,733 SNPs/InDels in 649 candidate genes and QTLs detected for agronomic traits and drought indices investigated under non-stressed (NS) and water-stressed (WS) environments. The ten sugarcane chromosomes (Sh01-Sh10) are shown as vertical bars, and each horizontal line on the bar represents SNPs/InDels in the candidate genes. The scale on the left side represents physical map positions in Mbp. Marker trait associations (MTAs) represented by blue, violet, brown, and red were detected for agronomic traits in NS-plant cane, NS-ratoon cane, WS-plant cane, and WS-ratoon cane, respectively, and MTAs represented by black and green were detected for indices in plant cane and ratoon cane, respectively. CL = cane length; CD = cane diameter; IL = internode length; SCW = single cane weight; NMC = number of millable canes; CY = cane yield; CCS = commercial cane sugar; FB = fiber; BR = brix; PO = polarization; PU = purity; SY = sugar yield; DC = the drought-tolerant coefficient; MFVD = the membership function value of drought tolerance.

**Figure 3 ijms-24-12801-f003:**
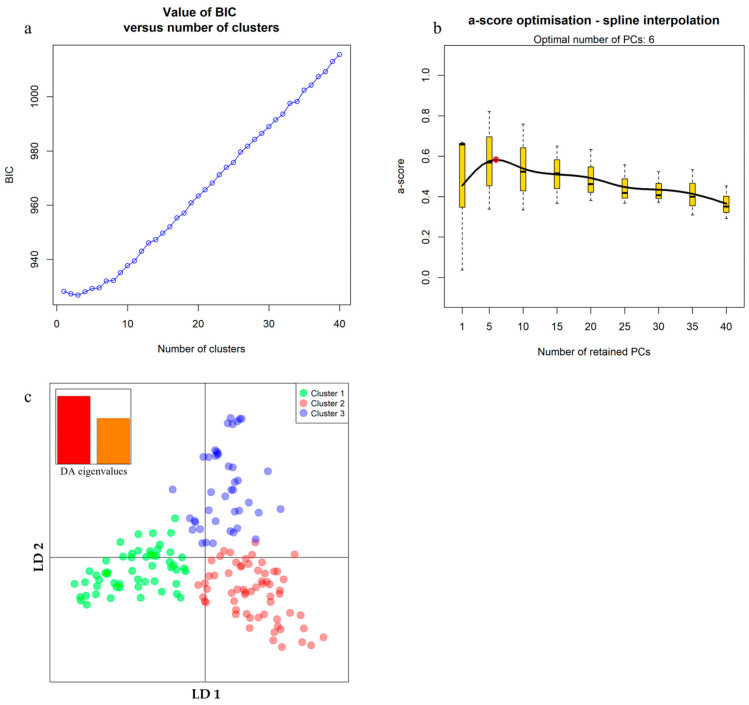
Population structure of the 159 sugarcane accessions based on Discriminant Analysis of Principal Components (DAPC) analysis on 5053 SNP-based markers. (**a**) BIC value shows the lowest at a cluster of three. (**b**) Several principal components (PCs) will be retained according to the maximized a-score at 6 PCs. (**c**) Scatter plot of first two linear discriminants (LD), representing accessions belonging to Cluster 1 (green group), Cluster 2 (red group), and Cluster 3 (blue group).

**Table 1 ijms-24-12801-t001:** Pearson’s correlation coefficients among the phenotypic traits in plant cane evaluated under the non-stressed (NS) experiment (the upper diagonal), the water-stressed (WS) experiment (the lower diagonal) and between the phenotypic traits evaluated under the NS and WS experiments (the diagonal).

	Traits	Non-Stressed											
		CL	CD	IL	SCW	NMC	CY	CCS	FB	BR	PO	PU	SY
Water Stressed	CL	0.60 **	−0.06	0.70 **	0.26 **	0.11	0.35 **	−0.36 **	0.36 **	−0.20 *	−0.33 **	−0.38 **	−0.06
	CD	−0.05	0.82 **	−0.26 **	0.90 **	−0.83 **	0.45 **	0.44 **	−0.68 **	0.04	0.34 **	0.51 **	0.58 **
	IL	0.71 **	−0.36 **	0.63 **	−0.03	0.28 **	0.26 **	−0.36 **	0.43 **	−0.19 *	−0.31 **	−0.39 **	−0.11
	SCW	0.19 *	0.77 **	−0.29 **	0.72 **	−0.74 **	0.58 **	0.31 **	−0.52 **	−0.01	0.23 **	0.38 **	0.57 **
	NMC	0.45 **	−0.55 **	0.59 **	−0.41 **	0.78 **	−0.14	−0.49 **	0.70 **	−0.08	−0.39 **	−0.56 **	−0.40 **
	CY	0.49 **	0.26 **	0.25 **	0.35 **	0.50 **	0.35 **	0.01	−0.25 **	−0.18 *	−0.05	0.07	0.60 **
	CCS	−0.48 **	0.28 **	−0.49 **	0.15	−0.50 **	−0.26 **	0.82 **	−0.49 **	0.77 **	0.97 **	0.93 **	0.77 **
	FB	0.51 **	−0.60 **	0.69 **	−0.45 **	0.64 **	0.08	−0.61 **	0.89 **	0.01	−0.34 **	−0.60 **	−0.48 **
	BR	−0.34 **	−0.05	−0.16 *	−0.19 *	−0.19 *	−0.33 **	0.70 **	−0.08	0.69 **	0.86 **	0.56 **	0.51 **
	PO	−0.44 **	0.09	−0.38 **	−0.02	−0.36 **	−0.30 **	0.93 **	−0.38 **	0.83 **	0.74 **	0.87 **	0.72 **
	PU	−0.46 **	0.28 **	−0.53 **	0.20 *	−0.52 **	−0.22 **	0.92 **	−0.62 **	0.49 **	0.87 **	0.74 **	0.77 **
	SY	0.16 *	0.42 **	−0.10	0.44 **	0.12	0.77 **	0.39 **	−0.32 **	0.13	0.29 **	0.37 **	0.40 **

* and ** indicate significant at *p* < 0.05 and *p* < 0.01, respectively. CL = cane length; CD = cane diameter; IL = internode length; SCW = single cane weight; NMC = number of millable canes; CY = cane yield; CCS = commercial cane sugar; FB = fiber; BR = brix; PO = polarization; PU = purity; SY = sugar yield.

**Table 2 ijms-24-12801-t002:** Pearson’s correlation coefficients among the phenotypic traits in ratoon cane evaluated under the non-stressed (NS) experiment (the upper diagonal) and the water-stressed (WS) experiment (the lower diagonal) and between the phenotypic traits evaluated under the NS and WS experiments (the diagonal).

	Traits	Non-stressed												
		CL	CD	IL	SCW	NMC	CY	CCS	FB	BR	PO	PU	SY	RA
Water Stressed	CL	0.47 **	−0.40 **	0.73 **	0.02	0.39 **	0.37 **	−0.25 **	0.44 **	−0.15	−0.19 *	−0.24 **	0.08	0.30 **
	CD	−0.14	0.78 **	−0.36 **	0.78 **	−0.83 **	0.25 **	0.23 *	−0.59 **	−0.01	0.12	0.25 **	0.34 **	−0.22 *
	IL	0.63 **	−0.32 **	0.56 **	−0.11	0.35 **	0.20 *	−0.33 **	0.41 **	−0.24 **	−0.33 **	−0.39 **	−0.11	0.10
	SCW	0.43 **	0.59 **	−0.06	0.44 **	−0.74 **	0.57 **	0.20 *	−0.47 **	−0.02	0.10	0.22 *	0.55 **	0.08
	NMC	0.51 **	−0.11	0.43 **	0.08	0.43 **	−0.05	−0.36 **	0.60 **	−0.15	−0.28 **	−0.38 **	−0.31 **	0.31 **
	CY	0.54 **	0.25 **	0.36 **	0.43 **	0.77 **	0.23 *	−0.07	−0.20 *	−0.19 *	−0.14	−0.08	0.64 **	0.57 **
	CCS	−0.23 *	0.13	−0.30 **	0.10	−0.35 **	−0.23 *	0.62 **	−0.35 **	0.85 **	0.95 **	0.95 **	0.70 **	0.02
	FB	0.23 *	−0.39 **	0.29 **	−0.21 *	0.28 **	0.02	−0.56 **	0.62 **	−0.02	−0.19*	−0.36 **	−0.39 **	0.02
	BR	−0.01	−0.24 **	−0.02	−0.11	−0.12	−0.25 **	0.61 **	−0.04	0.41 **	0.93 **	0.78 **	0.51 **	0.04
	PO	−0.15	0.03	−0.17	0.07	−0.28 **	−0.22 *	0.88 **	−0.41 **	0.77 **	0.48 **	0.94 **	0.62 **	0.02
	PU	−0.20 *	0.20 *	−0.25 **	0.17	−0.33 **	−0.18 *	0.86 **	−0.56 **	0.48 **	0.92 **	0.53 **	0.66 **	−0.01
	SY	0.35 **	0.31 **	0.17	0.44 **	0.47 **	0.80 **	0.29 **	−0.30 **	0.06	0.26 **	0.31 **	0.15	0.43 **
	RA	0.37 **	0.20 *	0.30 **	0.24 **	0.57 **	0.73 **	−0.11	−0.11	−0.16	−0.11	−0.07	0.61 **	0.07

* and ** indicate significant at *p* < 0.05 and *p* < 0.01, respectively. CL = cane length; CD = cane diameter; IL = internode length; SCW = single cane weight; NMC = number of millable canes; CY = cane yield; CCS = commercial cane sugar; FB = fiber; BR = brix; PO = polarization; PU = purity; SY = sugar yield; RA = ratooning ability.

**Table 3 ijms-24-12801-t003:** Descriptive statistic for drought tolerance indices evaluated in plant cane and ratoon cane. DC = Drought-tolerant coefficient of each yield trait; MFVD = the average value of the membership function of all the traits.

Trait	Mean	SD	Min	Max	CV (%)
Plant cane					
DC_CL_	0.78	0.09	0.54	1.15	14.03
DC_CD_	0.99	0.10	0.75	1.32	11.23
DC_IL_	0.88	0.13	0.58	1.32	13.93
DC_SCW_	0.71	0.19	0.25	1.62	21.65
DC_NMC_	0.86	0.26	0.35	1.83	15.80
DC_CY_	0.64	0.26	0.11	1.54	21.74
DC_SY_	0.67	0.30	0.10	1.78	25.34
MFVD	0.37	0.11	0.13	0.70	17.10
Ratoon cane					
DC_CL_	0.61	0.11	0.33	0.97	14.69
DC_CD_	0.84	0.10	0.56	1.12	9.43
DC_IL_	0.64	0.11	0.40	0.90	14.93
DC_SCW_	0.63	0.28	0.20	1.71	27.15
DC_NMC_	0.71	0.36	0.13	1.70	28.29
DC_CY_	0.51	0.30	0.12	1.45	30.86
DC_SY_	0.40	0.29	0.09	1.88	33.60
MFVD	0.36	0.13	0.11	0.67	23.19

**Table 4 ijms-24-12801-t004:** Pearson’s correlation coefficients between MFVD and DC of each yield trait evaluated in plant cane (the upper diagonal) and ratoon cane (the lower diagonal) and between the phenotypic traits evaluated under the non-stressed and water-stressed experiments (the diagonal). * and ** indicate significant at *p* < 0.05 and *p* < 0.01, respectively. DC = Drought-tolerant coefficient of each yield trait; MFVD = the average value of the membership function of all the traits.

	Traits	Plant Cane							
		DC_CL_	DC_CD_	DC_IL_	DC_SCW_	DC_NMC_	DC_CY_	DC_SY_	MFVD
Ratoon Cane	DC_CL_	0.42 **	0.36 **	0.55 **	0.46 **	0.21 *	0.51 **	0.34 **	0.71 **
	DC_CD_	0.15	0.14	0.29 **	0.40 **	0.18 *	0.42 **	0.29 **	0.63 **
	DC_IL_	0.59 **	0.14	0.40 **	−0.06	0.08	0.25 **	0.04	0.46 **
	DC_SCW_	0.60 **	0.28 **	0.30 **	0.29 **	0.14	0.40 **	0.32 **	0.53 **
	DC_NMC_	0.25 **	0.24 **	0.14	0.05	0.27 **	0.73 **	0.68 **	0.67 **
	DC_CY_	0.43 **	0.34 **	0.24 **	0.38 **	0.79 **	0.37 **	0.84 **	0.90 **
	DC_SY_	0.41 **	0.18 *	0.22 *	0.31 **	0.73 **	0.85 **	0.24 **	0.76 **
	MFVD	0.71 **	0.48 **	0.56 **	0.60 **	0.70 **	0.87 **	0.79 **	0.35 **

## Data Availability

Data is contained within the article or Appendix A.

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
