# Peer review of "A Large-Scale Candidate-Gene Association Mapping for Drought Tolerance and Agronomic Traits in Sugarcane"

_ijms, 2023, doi:10.3390/ijms241612801_

Round 1
Reviewer 1 Report
Comments to the author:
Comments for the Manuscript ID: ijms-2534369
The manuscript entitled "A Large-Scale Candidate-Gene Association Mapping for Drought Tolerance and Agronomic Traits in Sugarcane" submitted to "International Journal of Molecular Sciences" under the ‘Section: Molecular Plant Sciences” and Special Issue: Abiotic Stress Tolerance and Genetic Diversity in Plants is suitable for publication after minor revisions. The authors have identified 197 significant marker trait associations (MTAs) in 141 candidate genes 21 associated with 18 evaluated traits with the Bonferroni correction threshold (α=0.05). They mentioned 95 MTAs in 78 candidate genes and 62 MTAs in 58 candidate genes were detected under NS and WS conditions, respectively. The authors also identified 19 pleiotropic genes associated with more than one trait and many genes associated with drought tolerance indices. The experimental part is well designed and the data analysis is credible.
The specific comment:
The author must reduce the number of references and also remove the references before 2010.
English is okay but it will be very good if you check again.
Reviewer 2 Report
Dear authors,
The manuscript titled “A large-Scale Candidate-Gene Association Mapping for Drought Tolerance and Agronomic Traits in Sugarcane” provides an interesting perspective to explore the GWAS analysis in order to understand the genetic architecture of the Drought Tolerance and Agronomic Traits in Sugarcane. The authors found a total of 197 significant marker trait associations (MTAs) in 141 candidate genes associated with 18 evaluated traits. These results are very interesting in Sugarcane breeding plans, as well as to understand the genetic basis of physiological processes of Drought Tolerance.
In general, the authors have a logical sequence of analysis, in addition to having an extensive and variable panel of genotypes, as well as a high number of SNP molecular markers. However, it is strongly recommended to be more descriptive in the statistical methodology, in other words, the name of the algorithms implemented, the main parameters, and their references.
The above in order to have more clarity on how the analyzes were carried out, and thus be more effective in the reproducibility of the results in subsequent works. For all the above reasons, I recommend this manuscript for publication in IJMS with minor revision. However, the authors should address the following recommendations to improve some aspects before the publication (attached PDF document)
Best regards,

Reviewer 3 Report
This study provides a large-scale of genotype and phenotype dataset for drought tolerance in sugarcane. I feel Introduction lacks the explanation of the specific word in this study, such as MFVD, plant cane and ratoon cane. This lacking make Result session difficult for me. I did not clearly understand the purpose of the comparison of plant cane and ratoon cane.
I also feel this manuscript is too descriptive. Please clarify the new finding or new concept from obtained data set.
Comments
P1 L38 it is usually harvested 12 months after 38 planting.
Some references are required for the readers not familiar with Sugarcane cultivation.
P2 L62 The membership function value of drought tolerance (MFVD)
I am not familiar with the method, MFVD. Can you introduce the concept of this in the introduction? Such information will help readers to understand the results.
P2 L94 plant cane and ratoon cane
I am not familiar with the word, plant cane and ratoon cane. Please add the brief introductions of these.
P3 L127 the 159 sugarcane clones
I was confused about what “clones” mean. I feel 159 clones mean 159 clonal plants of the same genotype. Related to this point, the number of accessions used in this study should be clarified in the Abstract or Introduction. In addition, the detailed information of the germplasm will be helpful for the readers.
Figure 3 Panel C
I understand that this study revealed the three clusters. I wonder that this cluster, generated from the exogenic region sequences of 649 candidate gene, are similar to the results using neutral markers. At P11 L427, the effect of linkage background is discussed. Can you make new figure of the PCA with the plot color corresponding to such genetic background.
Reviewer 4 Report
Please find attached document

Round 2
Reviewer 3 Report
The authors have carefully revised the manuscript according to my comments.
Reviewer 4 Report
Dear authors,
You managed to replay to comments and suggestions.
All the best in your future scientific work.